# Air Temperature as a Key Indicator of Avocado (Cvs. Fuerte, Zutano, Hass) Maturation Time in Mediterranean Climate Areas: The Case of Western Crete in Greece

Thiresia-Teresa Tzatzani [1], Giasemi Morianou [2], Safiye Tül [1] and Nektarios N. Kourgialas [2,*]

[1] Subtropical Plants & Tissue Culture Laboratory/Institute of Olive Tree, Subtropical Crops and Viticulture, Hellenic Agricultural Organization (ELGO Dimitra), P.C. 73134 Chania, Greece
[2] Water Recourses-Irrigation & Environmental Geoinformatics Laboratory/ Institute of Olive Tree, Subtropical Crops and Viticulture, Hellenic Agricultural Organization (ELGO Dimitra), P.C. 73134 Chania, Greece
* Correspondence: kourgialas@elgo.gr; Tel.: +30-2821083442

**Abstract:** Avocado (*Persea americana* Mill.) is a tropical fruit that has gained immense popularity worldwide due to its unique flavor, nutritional value, and numerous health benefits. The growth and maturation of avocados are influenced by various environmental and climatic factors. Temperature is one of the most critical factors, as it plays a significant role in various physiological processes of this fruit. This study aimed to investigate the effect of air temperature on avocado development and assess the harvest maturity period (determined by the dry matter content) of Fuerte, Zutano, and Hass, the most common avocado varieties, grown in Western Crete, Greece. Fruit samples collected from avocado orchards in five regions of Western Crete were subjected to dry matter analysis during the ripening period for the years 2017 to 2022. Maturation time is determined based on the EU regulation for dry matter concentration. The results of this study revealed that dry matter concentration can be affected by both very high summer temperatures and low temperatures during the ripening period. Specifically, high temperatures during the summer months can have a significant impact on avocado development and result in a reduced dry matter concentration. On the other hand, low temperatures during the maturation stage can slow down enzymatic activity and metabolic processes, resulting in delayed ripening and a prolonged time to reach optimal maturity. This delay in maturation and reduced dry matter concentration can significantly affect the harvest timing, as growers may need to wait for the avocados to develop the desired characteristics before they can be harvested. Our findings are the first data on avocado maturation (dry matter concentration) in relation to temperature and provide valuable insights into harvest maturity period of avocado cultivation in Crete and other regions with similar Mediterranean climatic conditions.

**Keywords:** avocado; harvest maturity; subtropics; dry matter; temperature; Mediterranean region

## 1. Introduction

Avocado (*Persea americana* Mill.) is one of the most important socioeconomic tree crops in tropical and sub-tropical regions and has adapted to tropical and sub-tropical climate conditions relatively well [1]. Over the past few decades, its cultivation has expanded beyond traditional tropical and subtropical regions, and avocado production is now being explored in diverse climates and geographies. According to the FAO [2], the area devoted to avocado cultivation across the word reached 858,152 ha in 2021 and the production exceeded 8.5 million tons, 47% higher than it was five years ago. Spain is the leading avocado producer in the European Union, with more than 18,000 ha cultivated in 2021 [2]. In Greece, 1900 ha are cultivated, and 12,760 tons of avocados were produced during 2021. The main avocado production in Greece takes place in the western part of the island of Crete due to the region's suitable climate and soil conditions. As farmers and consumers have become more aware of its nutritional values, the popularity and production levels

of avocado have increased over the last decade in Crete [3]. The most common varieties are Hass, Fuerte, and Zutano. However, the successful cultivation of avocados in non-traditional regions requires a comprehensive understanding of the crop's response to various environmental factors.

The adaptability of the avocado crop allows it to thrive in tropical and subtropical climates, enabling its cultivation even in regions with a typical Mediterranean climate [4]. However, these areas often experience limited water availability during summer and face the risk of high salinity in irrigation water, especially in coastal regions [5]. Numerous studies have indicated that water scarcity is the primary restriction for crop production in the arid and semi-arid regions of the Mediterranean basin [6]. However, water scarcity in tree crops can be controlled by the producers, to a degree, through irrigation and cultivation practices [7]. On the other hand, other environmental factors that influence plant growth and development, such as air temperature, are not as easy to control for the producer. Air temperature affects various physiological processes, such as photosynthesis, respiration, and reproductive development [8,9]. Avocados are affected by temperature in several ways, including their flowering time, fruit development, ripening, yield, and fruit quality [10–12]. Avocados ripen at different times each year and are known to be affected by climatic conditions [13]. Thus, as global temperatures continue to rise due to climate change, it is becoming imperative to assess the effect of temperature on avocado development in regions previously unexplored for avocado cultivation, so that agricultural production systems can adapt to meet the rising demand for this food while coping with increasingly adverse weather conditions and the higher occurrence of extreme weather events.

Avocado fruit does not ripen on the tree but ripens after harvest; therefore, it is important to accurately determine the fruit's maturity. Otherwise, significant losses in avocado postharvest quality may occur. In both cases, immature and over-mature fruit harvests result in a shorter fruit shelf life, causing considerable economic losses. In addition, the harvesting of avocado fruits before they reach optimum maturity results in decreased levels of dry matter and a watery taste of the fruit [14,15]. On the other hand, the harvesting of over-ripe fruit can cause physiological disorders such as breakage of the fruit peel, degeneration of the fruit flesh, and a short shelf life [16]. Thus, the identification of the avocado's harvest maturity period is crucial.

However, identifying the harvest maturity of avocados can be challenging due to the absence of visible changes throughout the ripening process. While skin color and firmness can provide useful cues for ripeness in various fruits, in avocado, ripeness is primarily determined by measuring the dry matter concentration and oil content within the fruit. These internal factors indicate the fruit's maturity level and readiness for harvest and consumption. However, it is important to note that certain avocado varieties, such as Hass and Lamb Hass, exhibit a color change as they ripen. These varieties transition from green to a distinctive black-purple hue, providing an additional visual cue with which to assess their maturity. Dry matter concentration serves as a reliable maturity indicator in the assessment of avocados, since as the fruit develops, the dry matter and oil content also gradually increase [17]. More specifically, Clark et al. [18] and Woolf et al. [19] reported that a gradual decrease in mesocarp moisture content and an increase in dry matter content characterizes avocado fruit maturity. Dry matter represents the percentage of solid matter remaining after the removal of water, reflecting the composition of carbohydrates, fats, proteins, and other essential components within the fruit. The increase in dry matter concentration typically indicates a higher content of desirable nutrients and contributes to the development of an optimal flavor, texture, and overall fruit quality [15]. Thus, the monitoring and measuring of the dry matter concentration provides valuable insights for growers and researchers in determining the appropriate harvest timing, ensuring that the avocados have reached the desired maturity stage with enhanced nutritional value and consumer appeal [14,20]. Additionally, dry matter concentration serves as a vital parameter in post-harvest management and storage practices, aiding in the prediction of shelf life and maintaining fruit quality throughout the supply chain.

According to Gamble et al. [21] and Subedi et al. [22], avocado fruit can naturally remain on the tree for up to 12 months without ripening, resulting in higher dry matter concentration levels. However, an extended duration on the tree is associated with a shorter shelf life, increased vulnerability to disease, rancidity, and biennial bearing.

The harvesting time must be determined according to the EU Regulation 831/1997, as amended by Regulation 387/2005 [23], establishing the marketing standards applicable to avocados and the minimum dry matter concentration required to ensure the ripening process of avocado varieties. However, the market is putting pressure on Greek avocado producers to start harvesting in October, without always having carried out the necessary checks for the concentration of dry matter. The optimal harvesting time of avocado fruit that provides commercial flexibility to the market varies according to the cultivar, region, meteorological aspects, and other parameters; consequently, it is important to understand the different properties of each cultivar and to describe the maturity requirements [24–26].

This this study will contribute novel insights for avocado cultivation in Mediterranean climates, benefiting growers, agronomists, and researchers interested in expanding avocado production in similar environmental settings. Additionally, this research will enhance our knowledge of the complex relationship between air temperature and avocado growth and harvest maturity. The findings will support decision making in the avocado cultivation sector and promote sustainable avocado production in previously unexplored regions. Notably, this study represents one of the pioneering efforts to establish a correlation between avocado ripening (dry matter concentration) and temperature, offering valuable information on the harvest maturity period for avocado crops in Crete and other Mediterranean climate regions.

## 2. Materials and Methods

### 2.1. Study Area

The study was conducted from 2017 to 2022 in 10 avocado orchards of 9-year-old trees (mature avocado trees), located in the five most extensive avocado-growing areas in Western Crete (Vatolakkos, Alikianos, Mournies, Agrokipio, Apokoronas). The applied irrigation and fertilizing programs were the same for all the orchards. The island of Crete, located in the southernmost part of Greece, offers a unique environment for studying the response of avocado trees to ambient temperature. Crete includes four prefectures, the Prefectures of Chania and Rethymnon, located in the western part, and the Prefectures of Heraklion and Lassithi in the eastern part (Figure 1). The climate of Crete is characterized by hot, dry summers and mild, wet winters; thus, the island provides a representative Mediterranean environment in which to investigate the adaptability and productivity of avocados under varying temperature regimes. Furthermore, Crete encompasses lowland and highland areas, with altitudes ranging from 0 m to 2400 m above the mean sea level. Due to the complex topography of the island, there are significant climatological differences between the western and eastern parts of the island, with the western part of the island receiving higher amounts of precipitation compared to the eastern part [27,28].

Avocado, as a subtropical crop, has special climatic requirements in terms of ambient temperature and humidity (60–65%). It is also particularly sensitive to lack of water and excessive soil moisture. Regarding the soil, the important parameters for successful cultivation are (a) an appropriate pH of the soil, (b) a low percentage of carbonates ($CaCO_3$), (c) low salinity, and (d) good drainage [3]. For the above reasons, the avocado in Crete thrives mainly in the north-western part of Crete, especially in the north-central and north-eastern parts of the Chania Prefecture.

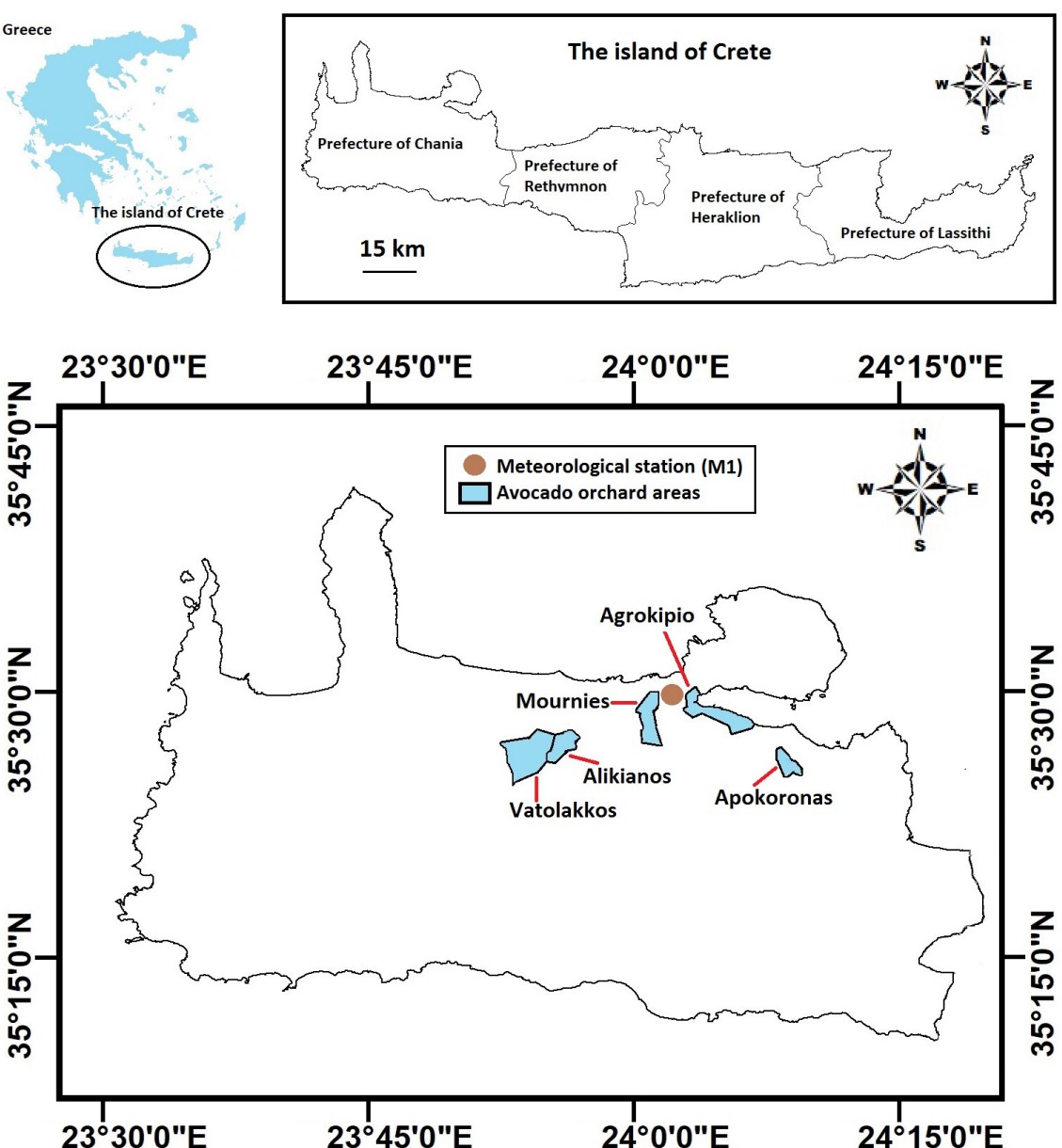

**Figure 1.** Avocado orchards of 9-year-old trees, located in five areas (communities) in Western Crete (Prefecture of Chania), named Vatolakkos, Alikianos, Mournies, Agrokipio, and Apokoronas (Datum: WGS 84).

*2.2. Weather Data Collection*

Indicating that variability in air temperature is the most important parameter, as compared to other meteorological parameters that affect various physiological processes of avocado trees, as well as the fruit development stages, this study highlights the relationship between air temperature and avocado harvest maturity period. Considering that the locations of the 10 studied avocado orchards are characterized by the same geomorphological and climate conditions, the meteorological parameters and, in particular, the temperature data were collected from the nearest and most representative meteorological station (M1) (0.5 and 10 km being the minimum and maximum distances from the avocado orchards, respectively), as depicted in Figure 1.

The mean monthly air temperature in the years 2017–2022 for the months of June to December are presented in Table 1. This period corresponds to the months of development and maturation of the three avocado varieties.

**Table 1.** Average monthly air temperature (°C) in the years 2017–2022 for the months of June to December (development and maturation stages).

| Year | June | July | August | September | October | November | December |
|------|------|------|--------|-----------|---------|----------|----------|
| 2017 | 23.8 | 26.7 | 25.9 | 23.5 | 18.4 | 15.3 | 13.5 |
| 2018 | 24.7 | 26.6 | 26.0 | 23.5 | 18.7 | 16.1 | 12.6 |
| 2019 | 18.4 | 25.3 | 25.2 | 22.8 | 20.1 | 18.1 | 14.1 |
| 2020 | 24.1 | 25.9 | 26.4 | 25.1 | 21.5 | 16.3 | 14.2 |
| 2021 | 24.3 | 28.0 | 28.1 | 24.2 | 19.4 | 18.0 | 13.7 |
| 2022 | 25.7 | 26.5 | 27.0 | 24.5 | 19.8 | 17.2 | 15.0 |

*2.3. Sampling and Lab Analysis*

In this study, samples were taken from the 10 avocado orchards once per week for dry matter analysis. The dry matter (DM) was measured for the Fuerte and Zutano varieties during the ripening period of 2017–2022 and for the Hass variety during ripening period of 2018–2022 (Figure 2).

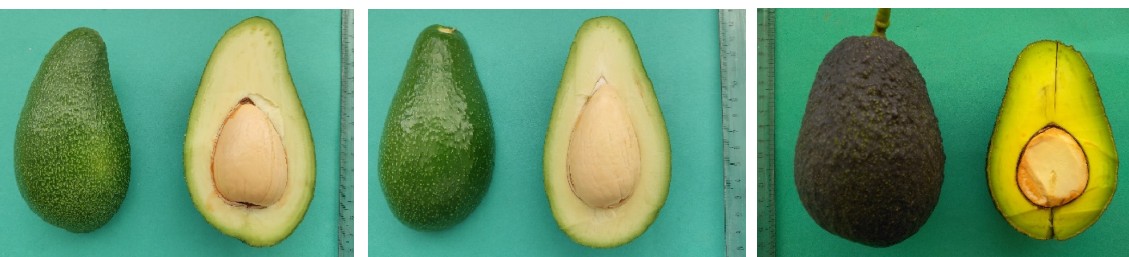

**Figure 2.** Fuerte, Zutano, and Hass avocado fruits (from left to right), ready for lab analysis.

Each sample consisted of at least five avocado fruits. The avocado fruits obtained from the farm were immediately taken to the lab (within an hour) for dry matter analysis. One fruit was kept at 20 °C for maturation (control). The remaining portion had the seed, seed coat, and skin removed. The fruit flesh was cut into smaller pieces and then blended into a paste using a mechanical blender. The fresh sample paste was placed in three preweighed ceramic pods, and the initial weight of each pod with the fresh sample was recorded. The samples were then dried by placing them in an oven at 103.5 °C overnight. After drying, the final weight of each sample was measured, and the dry matter content was calculated using Equation (1):

$$DM(\%) = \frac{DW}{FW} \times 100, \tag{1}$$

where DM is the dry matter (%), FW is the fresh weight (g), and DW is the dry weight (g).

*2.4. Statistical Analysis*

Differences in dry matter between the avocado fruits were analyzed using SPSS (SPSS Inc., Chicago, IL, USA) and subjected to one-way analysis of variance (ANOVA). The mean values were compared using Duncan's post hoc test for $p \leq 0.05$.

**3. Results**

Avocado fruit dry matter was measured to assess the effect of air temperature on avocado maturation time in Western Crete. The avocado varieties used in the experiment were the common ones cultivated in the area: Fuerte, Zutano, and Hass. The above varieties have distinct times of flowering, fruiting, and maturation that differ from year to year. However, in Crete, the exact harvest timed for the above varieties have not been determined. Producers in Crete start to harvest Fuerte and Zutano in October and Hass in December according to the market needs. Consequently, the sampling was carried out in the corresponding months for each variety. The EU regulation [21] defines the minimum

dry matter concentration required to ensure the ripening process for each variety: 21% for the Hass variety, 20% for Fuerte, and 19% for Zutano.

Figure 3 depicts the EU regulation thresholds [23] concerning the minimum dry matter concentration for harvesting as well as the range of dry matter (%) during the harvesting month from 2017 to 2022 for the Fuerte and Zutano varieties and from 2018 to 2022 for the Hass variety. The results of the dry matter analysis for the five years of the Fuente variety in October show that the dry matter value was only above 20% in the years 2018 and 2019; hence, the fruits were ready for harvest. The results for the rest of the years (2017, 2020–2022) were significantly lower (Figure 3a). For the Zutano variety, each of the October samples showed that for all the years (2017–2022), the concentration of dry matter was below the limit of 19% set by the EU regulation as an indicator of the maturation of the fruit. In particular, for the year 2017, the dry matter concentration in October was significantly lower compared to the years 2018–2022 (Figure 3b). Regarding the Hass variety, the dry matter in December was found to be higher than the threshold for harvest (21%) in every year, without any statistical difference between the years 2018 and 2022 (Figure 3c).

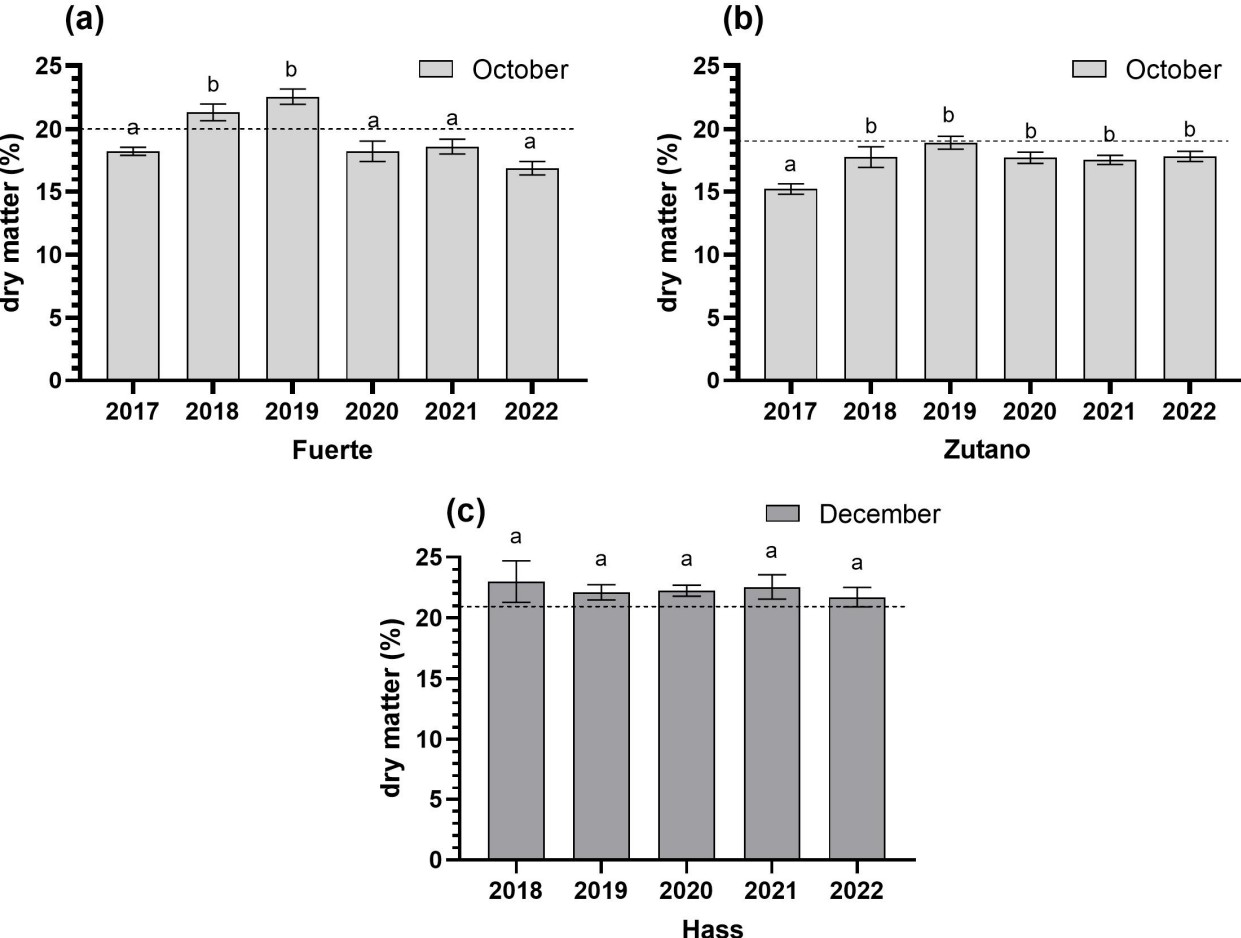

**Figure 3.** Dry matter concentration of the three studied avocado cultivars: (**a**) Fuerte, (**b**) Zutano, and (**c**) Hass. Comparison between the years 2017 and 2023 during the maturation month. Values are the means of the sampling replicates. Bars with the same letter are not significantly different at the 0.05 level of probability according to Duncan's test. The dashed line indicates the dry matter threshold for the maturation of each variety (20% for Fuerte, 19% for Zutano, and 21% for Hass).

Figure 4a shows the number of days during the months in which the fruit grew and matured from 2017 to 2022 on which the average air temperature exceeded 25 °C, and Figure 4b shows the number of days where the average air temperature was below 20 °C. The number of days on which the maximum air temperature exceeded 35 °C is

also shown in Figure 4c. The above thresholds of mean and maximum air temperature can be considered as representative temperature values for a hot-summer Mediterranean climate that cause stress for mature avocado trees, also depending on the duration of heat events [3,4].

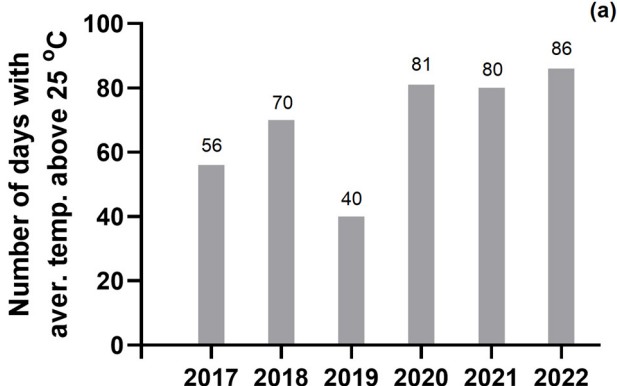

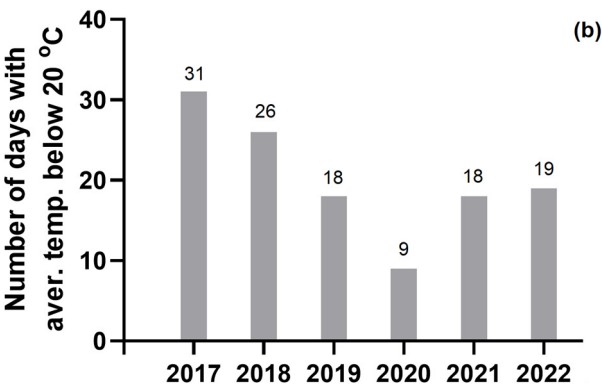

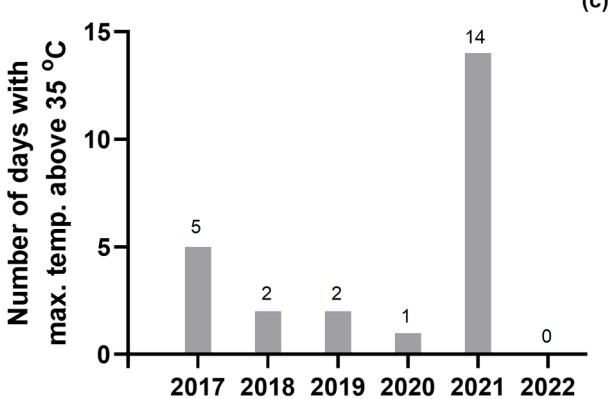

**Figure 4.** (**a**) Daily average temperatures recorded from June 2017 to December 2022 at Chania station, close to the studied orchards. The graph shows the number of days on which the average temperature exceeded 25 °C. (**b**) Daily average temperatures recorded from June 2017 to December 2022, showing the number of days on which the average temperature exceeded 25 °C. (**c**) Daily maximum temperatures recorded from June 2017 to December 2022 at Chania station. The graph shows the number of days on which the max. temperature exceeded 35 °C.

By analyzing the above temperature patterns and studying the dry matter contents, valuable insights can be derived. Specifically, in the years 2018 and 2019, when the Fuerte variety had high dry matter values and was ready for harvest, the average monthly temper-

atures in October were the highest compared to the other years (Table 1); this observation is in accordance with the results of Howden et al.'s [12] study, which indicates that warmer temperatures cause the fruits to set and mature sooner. Moreover, in 2019, the year with the smallest number of days with average temperatures above 25 °C during summer, (Figure 4a) the dry matter concentration of Fuerte was higher compared to that in 2018 and significantly higher compared to that in 2017 and 2020–2022. For the Zutano variety, this increase was also observed for 2019 in relation to the other years, albeit without a significant statistical difference (Figure 3b).

On the other hand, during 2017, when both the Fuerte and Zutano varieties had the lowest values of dry matter concentration, the average monthly temperature in October was the lowest across the 6 years (Table 1), while the average daily temperature never exceeded 20 °C (Figure 4b). In addition, an extreme heat event occurred during September (a critical month for both development and maturation), where temperatures reached 35 °C for more than 3 days.

It is also evident from the results of this study that in recent years (2020–2022), extreme climatic events, including frequent extreme heat events, significantly impaired the avocado maturation times of the Fuerte and Zutano varieties. For instance, during the summer of 2021, extreme heat waves occurred during all the summer months (Figure 4c), and the maximum daily air temperature in north-west Crete reached 43 °C. Although higher temperatures may favor quick maturation, extreme heat events seem to delay the dry matter increase of Fuerte and Zutano and thus delay their harvesting.

The results of this study show that both excessively high temperatures in summer and low temperatures during the ripening period can influence the concentration of dry matter in avocados. This research indicates that elevated temperatures during the summer season have a notable impact on avocado development, leading to a decrease in dry matter concentration. This can lead to avocados with lower concentrations of sugars, oils, and other desirable components, ultimately affecting their maturation process. Conversely, lower temperatures during the maturation stage can slow down enzymatic activity and metabolic processes, resulting in delayed ripening and an extended duration to reach optimal maturity. These delays in maturation and reduced dry matter concentration have significant effects on the harvest timing.

In addition, the results of this study show that in October, when the demand for Greek avocados in the market arises and producers are pressured to start harvesting, the Zutano variety was not ready for harvesting in all the studied years (2017–2022), and the Fuerte variety was not ready in 4 of the 6 years of the study. Avocado fruit harvesting time is crucial for the market and should occur at the appropriate time to obtain the desired quality [26].

The impact of temperature on dry matter concentration in avocados can vary depending on the cultivar and harvest timing. In this work, it was observed that summer and autumn temperatures during October affected the dry matter content of Fuerte and Zutano avocados but not Hass avocados harvested in wintertime. It seems that the ripening time of the Hass variety was not affected by the high summer temperatures and the differences in autumn temperatures.

These findings suggest that the temperature sensitivity of dry matter accumulation differs between avocado cultivars and their corresponding harvest months. This highlights the importance of considering varietal characteristics and harvest timings when assessing the relationship between temperature and dry matter concentration in avocados. Further research is needed to investigate the specific physiological mechanisms that contribute to these variations and to develop targeted temperature management strategies for different avocado cultivars. Additional research is also needed to determine whether climate change will impact specific parameters of avocado crops (i.e., the time between flowering and harvest) or will contribute to permanent changes in avocado cultivation (harvest time).

## 4. Conclusions

This study presented important insights into the relationship between air temperature and avocado maturation in Mediterranean climate regions. The concentration of dry matter during the maturation period was examined for the three most common varieties cultivated in the island of Crete (Fuerte, Zutano, and Hass) in the years 2017–2022. The results show that high temperatures during the summer and low temperatures during the maturation stage can have a significant impact on avocado maturation time in regard to the concentration of dry matter in the fruit. High temperatures during the summer months can accelerate the fruit's growth but may also result in increased water loss and reduced dry matter accumulation. In addition, low temperatures during the maturation stage can slow down enzymatic activity and metabolic processes, resulting in delayed ripening and a prolonged time to reach optimal maturity. This delay in maturation and reduced dry matter concentration can significantly affect harvest timing, as growers may need to wait for the avocados to develop the desired characteristics before they can be harvested.

Specifically, these analyses showed that in every year, the Zutano variety was not ready for harvest in October, since the dry matter concentration was below the EU regulation threshold of 19% in all the samples. Similarly, for the Fuerte variety, the dry matter analysis was above the threshold for harvest, being 20%, in October in only 2 of the 6 years. It appears that due to the region's climate, with high summer temperatures and possible low autumn temperatures, these two varieties are not ready for harvest in October, as the market demands. On the other hand, the Hass variety, which is harvested in the winter months, seems to be unaffected by the summer and autumn temperature differences. The dry matter values in the month of December were above the limit of 21%.

By understanding the relationship between air temperature and avocado maturation, farmers in Western Crete and similar areas can optimize their harvesting practices and improve their overall crop management. Furthermore, this knowledge can aid in the development of appropriate strategies and agricultural management practices for avocado cultivation, including the selection of suitable cultivars and the timing of harvests to maximize yield and fruit quality.

**Author Contributions:** Conceptualization, T.-T.T., G.M. and N.N.K.; lab-analysis T.-T.T. and S.T.; statistical analysis and data curation, G.M.; writing—original draft preparation, T.-T.T. and G.M.; writing—review and editing, N.N.K.; supervision, T.-T.T. and N.N.K. All authors have read and agreed to the published version of the manuscript.

**Funding:** This research received no external funding.

**Institutional Review Board Statement:** Not applicable.

**Data Availability Statement:** Data available on request due to restrictions (privacy).

**Conflicts of Interest:** The authors declare no conflict of interest.

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
