# Peer review of "Air Temperature as a Key Indicator of Avocado (Cvs. Fuerte, Zutano, Hass) Maturation Time in Mediterranean Climate Areas: The Case of Western Crete in Greece"

_agriculture, doi:10.3390/agriculture13071342_

Round 1
Reviewer 1 Report
The article is innovative and offers the possibility to undesrtand the relation between air temperature and avocado developmentand harvest maturity period. It needs a minor review.

Author Response
|
Page |
Line |
Comments |
Answers |
|
1 |
15 |
Western Crete – Greece |
Ok, please see line 15 |
|
1 |
16 |
Western Crete |
Ok, please see line 16 |
|
4 |
Fig. 1 |
Insert in Figure 1 title, legend, cartographic scale, orientation and projection. |
Ok, please see Fig. 1 |
|
4 |
Table 1 |
You can reduce the font letter or put the reduced name of the months in order to fit better in the table. |
Ok, please see Table 1 |
|
5 |
155 |
Insert a blank line below the paragraph |
Ok |
|
5 |
165 |
… according to the following Equation 1. |
Ok, please see line 181 |
|
5 |
165-166 |
Put (Eq. 1) right aligned |
The template of the paper/journal indicates the form (1) |
|
6 |
Fig. 3 |
I suggest to put the figure after the paragraph referent to it |
Ok |
|
7 |
Fig. 4 |
Same recommendation above |
Ok |

Reviewer 2 Report
Although the results are interesting for this field, I believe they are still too preliminary. A more in-depth study is needed.
- The manuscript as a whole is unnecessarily long and requires concise and focused revision. Certain parts, such as lines 172-183, are repetitive and duplicate information already presented in the introduction.
- Pleas specify the distance between each orchard to the weather station and explain as why only one weather station is sufficient enough to represent the real time temperature of all 10 orchards
- Regarding Table 2 and Figure 3, both contain the same result data. It is advisable to select only one type, either a table or a figure, to avoid redundancy.
- The study does not clearly establish the extent to which high or low factors impact DM in avocados, nor does it provide clear information on how long these factors affect fruit development or delay flowering. Furthermore, there is no evidence supporting the claim that keeping fruit on the tree longer during October harvest will result in higher DM levels, even for varieties like Zutano or Fuerte.
While the content is informative and valuable to the field, there are instances where the text appears to be redundant. Thus the author should enhance the clarity with a focus on eliminating unnecessary repetitions and ensuring a more concise presentation of their ideas.
Author Response
Reviewer 2
Comments and Suggestions for Authors
Although the results are interesting for this field, I believe they are still too preliminary. A more in-depth study is needed.
- The manuscript as a whole is unnecessarily long and requires concise and focused revision. Certain parts, such as lines 172-183, are repetitive and duplicate information already presented in the introduction.
To avoid repetition between Introduction (lines 92-97) and Results (lines 172-183), changes were made at the Introduction section, please see lines 103-105.
- Please specify the distance between each orchard to the weather station and explain as why only one weather station is sufficient enough to represent the real time temperature of all 10 orchards.
Your comment was taken into consideration, please see lines 157-159
- Regarding Table 2and Figure 3, both contain the same result data. It is advisable to select only one type, either a table or a figure, to avoid redundancy.
Your comment was taken into consideration, thank you.
- The study does not clearly establish the extent to which high or low factors impact DM in avocados, nor does it provide clear information on how long these factors affect fruit development or delay flowering. Furthermore, there is no evidence supporting the claim that keeping fruit on the tree longer during October harvest will result in higher DM levels, even for varieties like Zutano or Fuerte.
It is not possible to quantify the effect of temperature in such detail (i.e., 10 days of heat resulted in 10 days of delayed development or flowering) because there are other factors that may affect development and ripening too. In this work, an attempt is made for the first time to correlate the temperature (and extreme heat events) with the ripening time.
There are many studies in the literature indicating that avocado fruit can naturally remain on the tree for up to 12 months without ripening, resulting in higher dry matter concentration levels, please see the added line 98-101 in the Introduction section.
Comments on the Quality of English Language
- While the content is informative and valuable to the field, there are instances where the text appears to be redundant. Thus, the author should enhance the clarity with a focus on eliminating unnecessary repetitions and ensuring a more concise presentation of their ideas.
Your comment was taken into consideration, thank you.

Reviewer 3 Report
The manuscript, entitled " Air temperature as a key indicator of Avocado (Cvs. Fuerte, Zutano, Hass) maturation time in Mediterranean climate areas: The case of Western Crete in Greece " aims to investigate the effect of air temperature on avocado development and assess the harvest maturity period (determined by dry matter content) of Fuerte, Zutano and Hass, the most common avocado varieties, grown in western Crete – Greece. It found that on the one hand, high temperatures in summer can have a significant impact on the development of avocados and lead to a decrease in dry matter concentration. On the other hand, low temperatures during the maturation stage can slow down enzyme activity and metabolic processes, leading to delayed maturation and an extension of the time to reach optimal maturity. Although studies like this are valuable in application, the studies are not novel. I suggest exploring the correlation between the size of fruit kernels and fruit maturity in the discussion section. Some details should be improved, such as page 5, line 160, 164 ... the temperature unit is inconsistent with the image and other positions.
Author Response
Reviewer 3
Comments and Suggestions for Authors
The manuscript, entitled " Air temperature as a key indicator of Avocado (Cvs. Fuerte, Zutano, Hass) maturation time in Mediterranean climate areas: The case of Western Crete in Greece " aims to investigate the effect of air temperature on avocado development and assess the harvest maturity period (determined by dry matter content) of Fuerte, Zutano and Hass, the most common avocado varieties, grown in western Crete – Greece. It found that on the one hand, high temperatures in summer can have a significant impact on the development of avocados and lead to a decrease in dry matter concentration. On the other hand, low temperatures during the maturation stage can slow down enzyme activity and metabolic processes, leading to delayed maturation and an extension of the time to reach optimal maturity. Although studies like this are valuable in application, the studies are not novel. I suggest exploring the correlation between the size of fruit kernels and fruit maturity in the discussion section. Some details should be improved, such as page 5, line 160, 164 ... the temperature unit is inconsistent with the image and other positions.
Although studies like this are valuable in application, the studies are not novel:
Please see the reference to the novelty of the work in the lines 112-121 and 312-317
I suggest exploring the correlation between the size of fruit kernels and fruit maturity in the discussion section:
The correlation between the size of fruit kernels and fruit maturity in avocados is not a widely studied topic. The most common method used to assess avocado maturity is the dry matter content and the oil concentration analysis. The size of the fruit kernels, also known as the seeds, is generally not considered a reliable indicator of avocado maturity. The seed size is primarily influenced by genetic factors and may not necessarily correlate with the overall maturity of the fruit.
We also added lines 76-82 at the introduction section.
Some details should be improved, such as page 5, line 160 -164:
Thank you for the comment, this paragraph improved, please see lines 175-181.
The temperature unit is inconsistent with the image and other positions:
Both in figures and text the temperature unit is oC.

Round 2
Reviewer 2 Report
The responses and revision made by the authors is satisfied.